# Heparin Inhibits SARS-CoV-2 Replication in Human Nasal Epithelial Cells

**DOI:** 10.3390/v14122620

**Published:** 2022-11-24

**Authors:** Leo Yi Yang Lee, Randy Suryadinata, Conor McCafferty, Vera Ignjatovic, Damian F. J. Purcell, Phil Robinson, Craig J. Morton, Michael W. Parker, Gary P. Anderson, Paul Monagle, Kanta Subbarao, Jessica A. Neil

**Affiliations:** 1Department of Microbiology and Immunology, The University of Melbourne at the Peter Doherty Institute for Infection and Immunity, Melbourne, VIC 3000, Australia; 2Department of Respiratory Medicine, Royal Children’s Hospital, Parkville, VIC 3052, Australia; 3Infection and Immunity, Murdoch Children’s Research Institute, Royal Children’s Hospital, Parkville, VIC 3052, Australia; 4Department of Paediatrics, The University of Melbourne, Parkville, VIC 3052, Australia; 5Haematology, Murdoch Children’s Research Institute, Parkville, VIC 3052, Australia; 6Department of Biochemistry and Pharmacology, Bio21 Molecular Science and Biotechnology Institute, The University of Melbourne, Parkville, VIC 3010, Australia; 7St. Vincent’s Institute of Medical Research, Fitzroy, VIC 3065, Australia; 8Lung Health Research Centre, Department of Biochemistry and Pharmacology, The University of Melbourne, Parkville, VIC 3010, Australia; 9Department of Haematology, Royal Children’s Hospital, Parkville, VIC 3052, Australia; 10Kids Cancer Centre, Sydney Children’s Hospital, Randwick, NSW 2031, Australia; 11WHO Collaborating Centre for Reference and Research on Influenza, Melbourne, VIC 3000, Australia

**Keywords:** SARS-CoV-2, heparin, antiviral, nasal epithelial cells

## Abstract

SARS-CoV-2 is the causative agent of the COVID-19 pandemic. Vaccination, supported by social and public health measures, has proven efficacious for reducing disease severity and virus spread. However, the emergence of highly transmissible viral variants that escape prior immunity highlights the need for additional mitigation approaches. Heparin binds the SARS-CoV-2 spike protein and can inhibit virus entry and replication in susceptible human cell lines and bronchial epithelial cells. Primary infection predominantly occurs via the nasal epithelium, but the nasal cell biology of SARS-CoV-2 is not well studied. We hypothesized that prophylactic intranasal administration of heparin may provide strain-agnostic protection for household contacts or those in high-risk settings against SARS-CoV-2 infection. Therefore, we investigated the ability of heparin to inhibit SARS-CoV-2 infection and replication in differentiated human nasal epithelial cells and showed that prolonged exposure to heparin inhibits virus infection. Furthermore, we establish a method for PCR detection of SARS-CoV-2 viral genomes in heparin-treated samples that can be adapted for the detection of viruses in clinical studies.

## 1. Introduction

COVID-19 has caused significant morbidity and mortality worldwide since the emergence of its causative agent, the beta-coronavirus SARS-CoV-2, in late 2019 [1]. Importantly, the execution of social and preventive measures and the rapid development and implementation of many efficacious vaccines have curbed virus spread, reduced disease severity and saved approximately 20 million lives [2,3,4]. However, the continued emergence of SARS-CoV-2 variants with the ability to escape vaccine immunity, leading to breakthrough infections, highlights the need for broadly targeted strategies to block virus spread [5]. Furthermore, after an initial increase in IgA, current vaccine technologies do not induce sustained protective antibody responses in the nasopharynx, enabling recurrent infections [6]. There are only three licensed antiviral drugs currently in use against SARS-CoV-2 infection. This includes two that inhibit the SARS-CoV-2 viral polymerase (remdesivir and molnupiravir) and a protease inhibitor (Paxlovid) [7,8,9]. The use of these antivirals is often restricted to high-risk groups due to the cost of their administration. Inexpensive antiviral drugs that are effective in prophylaxis would be very valuable in preventing virus spread.

Targeting viral entry presents an appealing antiviral strategy. Although several drugs targeting host proteases and endosomal fusion factors involved with SARS-CoV-2 entry have had limited clinical success [10,11], antibodies targeting the SARS-CoV-2 entry receptor, angiotensin-converting enzyme-2 (ACE-2), have proved to be useful, though their utility is limited by the emergence of variant viruses with resistance mutations [12]. In 2020, it was shown that the spike (S) protein on the surface of the SARS-CoV-2 virus, required for binding to ACE2, also binds heparan sulfate, a complex polysaccharide belonging to the glycosaminoglycan family [13]. Importantly, SARS-CoV-2 entry was shown to be co-dependent on both ACE2 and heparan sulfate [13,14]. Neutralizing antibodies isolated from COVID-19 patients can also inhibit SARS-CoV-2 binding to heparan sulfate [15]. Thus, binding to heparan sulfate is likely important for SARS-CoV-2 infection in humans. 

Heparin is a polysaccharide that is produced by mast cells and is structurally similar to heparan sulfate [16]. Claussen and colleagues reported that heparin can bind to the SARS-CoV-2 virus S protein and block virus replication in Vero cells and human bronchial epithelial cells [13]. Unfractionated heparin and heparin derivatives have also been shown to block SARS-CoV-2 and SARS-CoV-2 pseudovirus infection in cell lines, such as Huh 7.5, Caco-2, Calu-3, HEK-293T and Vero E6 [15,17,18]. In addition to allosteric inhibition of the interaction between ACE2 and heparan sulfate, binding of heparin may also prevent furin cleavage of the S protein [19]. These in vitro studies suggest that heparin may be useful as an antiviral against SARS-CoV-2. Heparin is a low-cost and widely available drug that has been safely administered parenterally to humans for the treatment of a variety of medical conditions as an anticoagulant and by high-dose aerosol inhalation to the lungs to treat acute respiratory distress syndrome-like lung injury in very severe COVID, making it an attractive drug for repurposing as an antiviral [20]. Aerosolized heparin delivered to the nasal mucosa targets the initial site of infection. Topically, applying heparin to the nasopharynx as an aerosol generates negligible systemic bioavailability and does not alter prothrombin time, a sensitive coagulation index (Clinical Trial NCT04490239) [21]. This approach should maximize the anti-SARS-CoV-2 therapeutic index by delivering high concentrations to the site of infection with minimal risk of potential systemic side effects.

We confirmed that heparin can bind to the SARS-CoV-2 S protein using computational modeling. To validate the use of intranasal heparin, we assessed the effectiveness of heparin at inhibiting SARS-CoV-2 infection in Vero cells and human nasal epithelial cells. We show that heparin can inhibit SARS-CoV-2 replication in both cell types, with almost complete reduction in infectious virus in human nasal epithelial cells. Importantly, our data indicate that prolonged heparin exposure in cell culture is required for effective inhibition of virus growth and that heparinase treatment of heparin-contaminated samples is an effective method to ensure appropriate detection of viral RNA.

## 2. Materials and Methods

### 2.1. Molecular Modeling

Model heparin—spike complexes were created by manual addition of three 31-mer heparin strands to the cryo-electron microscopy structure of the SARS-CoV-2 S protein PDB code 6ZGE [22]. A single heparin strand was added manually in SybylX2.1 (Certara, Princeton, NJ, USA) to follow the positively charged surface of one monomer of the spike protein structure, and then the strand was copied using the three-fold symmetry of the S protein to generate the trimeric heparin—spike protein complex. The model was minimized to remove minor clashes introduced during the construction process in SybylX2.1. For SARS-CoV-2 variants, sequence differences were either introduced through simple mutations of the residues in the existing models or, in the case of variants with significant insertions or deletions, through the construction of homology models of the S protein using MODELLER version 10.1 [23]. Heparin—spike protein complex models were inspected with PyMOL (the PyMOL Molecular Graphics System, Version 2.5 Schrödinger, LLC, New York, NY, USA).

### 2.2. Virus and Cells

Vero cells (ATCC CCL-81) were cultured in minimal essential medium (MEM, Media Preparation Unit, Peter Doherty Institute) supplemented with 50 units/mL penicillin (Thermo Fisher Scientific, Waltham, MA, USA), 50 μg/mL streptomycin (Thermo Fisher Scientific), 2 mM GlutaMax (Thermo Fisher Scientific), 15 mM HEPES (Thermo Fisher Scientific) and 5% fetal bovine serum (Bovogen, Keilor East, VIC, Australia). Vero hSLAM cells (Sigma-Aldrich, St. Louis, MO, USA) were cultured in MEM supplemented with 7% fetal bovine serum, 50 units/mL penicillin, 50 µg/mL streptomycin, 2 mM GlutaMAX, 15 mM HEPES and 0.4 mg/mL G418 Sulfate (Thermo Fisher Scientific). The ancestral SARS-CoV-2 (Australia/VIC01/2020) was a generous gift from the Victorian Infectious Disease Research Laboratory (VIDRL). GFP-tagged SARS-CoV-2 (icSARS-CoV-2-GFP/WA1) was a kind gift from Professor Ralph Baric (Department of Microbiology and Immunology at the University of North Carolina, Chapel Hill, NC, USA). To propagate SARS-CoV-2, inoculated Vero cells or Vero hSLAM cells were cultured until at least 80% cytopathic effect (CPE) was observed. The virus-containing supernatant was clarified by centrifugation and stored at −80 °C. The titer (median tissue culture infectious dose per mL; TCID_50_/mL) of stock viruses was determined by an infectivity assay using Vero cells, as detailed below. VIC01 was passaged four times in Vero cells, and the GFP virus was passaged once in Vero hSLAM cells to obtain stocks for all infection assays.

### 2.3. Human Nasal Airway Epithelial Cell Culture at Air-Liquid Interface (ALI)

Healthy adults were recruited for this study under ethics approval HREC/35132. This study included 4 donors (3M, 1F) aged between 26 and 67 years. Nasal epithelial cells were collected from behind the subjects’ inferior nasal turbinates using 3 mm cytology brush (Conmed, Largo, FL, USA) and agitated in Medium-199 pH 7.6 (Sigma-Aldrich) containing 200 units/mL penicillin, 200 µg/mL streptomycin and 0.5 µg/mL amphotericin B (Sigma-Aldrich) before washing with excess PBS and centrifugation at 600× *g* for 5 min. Cells were then seeded into a 12-well cell culture plate coated with 0.1 mg/mL Collagen I (Sigma-Aldrich) and cultured with PneumaCult™-Ex Plus medium (STEMCELL Technologies, Vancouver, BC, Canada) supplemented with 96 ng/mL of hydrocortisone (STEMCELL Technologies), 200 units/mL penicillin, 200 µg/mL streptomycin and 0.5 µg/mL amphotericin B. Cultures were expanded from the initial seeding using a collagen-coated T75 cell culture flask. For ALI culture, 3.6 × 10^5^ cells were seeded per well into the apical area of 12 mm Transwell^®^ with 0.4 μm pore polyester membrane insert and cultured with supplemented PneumaCult™-Ex Plus medium both apical and basolateral compartments. Once the basal cells formed a fully confluent monolayer, the cells were then exposed to ambient air with PneumaCult^TM^-ALI medium (STEMCELL Technologies) supplemented with 4 µg/mL heparin (STEMCELL Technologies), and 480 ng/mL hydrocortisone on the basolateral side only. PneumaCult^TM^-ALI medium was replaced 3 times a week, for 6 weeks where optimal ciliation was usually observed. 

### 2.4. SARS-CoV-2 Infection

Twenty-four-well plates of confluent Vero cells (approximately 5 × 10^5^ cells/well) were infected with 10^2^ (multiplicity of infection (MOI) 0.0002) TCID_50_ of SARS-CoV-2 for 1 h at 37 °C. Following removal of the inoculum, cells were washed with serum-free MEM (50 units/mL penicillin, 50 μg/mL streptomycin, 2 mM GlutaMax and 15 mM HEPES) and cultured for 3 days. Each day, the supernatant was collected and replaced with fresh serum-free MEM. Nasal ALI cells were infected apically with 10^4^ (approx. MOI 0.2) TCID_50_ of SARS-CoV-2 for 2 h at 37 °C. Following removal of the inoculum, the apical surface of ALI cells was washed three times with phosphate-buffered saline (PBS) for 10 min each. Cells were cultured for 7 days with basolateral media changes on days 0, 2, 4 and 6 post infection. Each day, 100 μL of PBS added to the apical surface for 30 min at 37 °C was collected for RT-PCR or virus titration. All work with infectious viruses was performed inside a biosafety II cabinet, in a biosafety containment level 3 facility, and personnel wore powered air-purifying respirators (3M TR-315A VERSAFLO) or P2 masks.

### 2.5. Heparin-Treatment

Vero and ALI cells were pretreated with either 125 µg or 250 μg of heparin sodium (Pfizer, New York, NY, USA) for 30 min prior to infection with SARS-CoV-2 as detailed above. Following infection (unless indicated otherwise), cells were cultured in the presence of either 125 μg/mL or 250 μg/mL of heparin throughout the experiment. For ALI nasal epithelium, cells were treated with heparin on both apical and basolateral surfaces, with heparin replaced daily on the apical surface and every 2 days on the basolateral surface. In some experiments, the virus was incubated with heparin (250 μg) for 1 h at room temperature prior to infection, as described above. The virus/heparin solution was removed from the cells by washing with either serum-free MEM (Vero) or PBS (ALI cells) and cultured with or without heparin for the remainder of the experiment. Cell viability in the presence of heparin was assessed using the CellTitre-Glo kit (Promega, Madison, WI, USA) as per the manufacturer’s instructions.

### 2.6. Virus Titration

TCID_50_ in supernatants was determined by titration on Vero cells. Briefly, confluent monolayers of Vero cells in 96-well plates were washed with MEM and replaced with 180 μL of infection media (serum-free MEM with 1μg/mL TPCK-treated Trypsin (Worthington Biochemical Corp, Lakewood, NJ, USA). Additionally, 20μL of supernatant was added to quadruplicate wells, and 10-fold serial dilutions were created. Plates were incubated at 37 °C and 5% CO_2_ for 4 days before being assessed for SARS-CoV-2-induced CPE by microscopy. Virus titers were calculated using the Reed and Muench method and expressed in TCID_50_/mL. The limit of detection for this infectious virus assay was 10^1.2^ TCID_50_/mL.

### 2.7. Viral RT-PCR

RNA was extracted from supernatants using the QIAamp Viral RNA Mini Kit (Qiagen, NRW, Hilden, Germany). To prevent the inhibitory effect of heparin during RT-PCR, extracted RNA (8 μL) was treated with 0.5 units of Heparinase I (Sigma-Aldrich, diluted in 20 mM Tris, 600 mM NaCl, 150 mM CaCl_2_, pH 7.0) and 40 units of RNAseOUT (Thermo Fisher Scientific) for 1 h at room temperature. Treated RNA was assessed for E-gene levels using the SensiFAST Probe No-Rox One Step Kit (Bioline, London, UK) and the following primers/probes: Fwd: 5′-ACAGGTACGTTAATAGTTAATAGCGT’-3, Rev: ATATTGCAGCAGTACGCACACA and Probe: FAM-ACACTAGCCATCCTTACTGCGCTTCG-BBQ. Viral genomes were interpolated using a standard curve generated by a plasmid containing the E-gene.

### 2.8. GFP Virus Detection

On days 1 and 3 post infection, Vero cells were detached using Trypsin-Versene (Media Preparation Unit, Peter Doherty Institute) and stained for 15 min at room temperature with Zombie NIR Fixable dye (Biolegend, San Diego, CA, USA). Cells were then fixed for 30 min with 4% (*v/v*) formaldehyde and run on a BD FACSCanto II using FACS Diva Software. Data was analyzed using FlowJo (BD, Franklin Lakes, NJ, USA). GFP expression was also observed visually using an EVOS M5000 cell imaging system (Thermo Fisher Scientific).

### 2.9. Immunofluorescence

Nasal epithelium on transwells were fixed with 4% formaldehyde solution for 30 min, washed three times with PBS and stored at 4 °C until staining. Transwells were whole mount stained by treating with blocking solution (5% normal goat serum, 0.1% Triton-X in PBS) for 1 h at room temperature and staining overnight at 4 ^º^C with mouse anti-dsRNA (J2, Mouse IgG2a kappa, Australian Biosearch, WA, Australia) and rabbit anti-SARS coronavirus nucleocapsid polyclonal antibody (PA1-41098, Thermo Fisher Scientific). After washing with PBS, primary antibodies were detected using Goat anti-Mouse IgG (H + L) Cross-Adsorbed Alexa Fluor 647 (Thermo Fisher Scientific) and Goat anti-Rabbit IgG (H + L) Cross-Adsorbed Alexa Fluor 568 (Thermo Fisher Scientific) for 1 h at room temperature. Cells were counterstained with DAPI (Thermo Fisher Scientific) and transwells mounted with VECTASHIELD Antifade Mounting Medium (Vector Laboratories, Newark, CA, USA) on SuperFrost slides (ProSciTech, Kirwan, QLD, Australia) before visualization using the LSM 780 (Ziess, BW, Oberkochen, Germany). 

### 2.10. Statistical Analysis

All data were plotted and analyzed using GraphPad Prism 9. Log_10_ virus titers were analyzed using either a Student’s *t*-test, One-Way ANOVA or Two-Way ANOVA with Dunnett’s multiple comparisons test as appropriate. *, *p* < 0.05; **, *p* < 0.01; ***, *p* < 0.001 and ****, *p* < 0.0001

## 3. Results

### 3.1. Modeling of SARS-CoV-2 Spike with Heparin

Inspection of the SARS-CoV-2 S protein structure identified potential polysaccharide binding surfaces that could provide a receptor surface for heparan sulfate. We modeled the omicron variant S protein—heparin complex with one strand of heparin bound to a groove in each of the trimeric S protein chains (Figure 1). The highly negatively charged surface of heparin interacts with a positively charged surface on the SARS-CoV-2 S protein. Importantly, the SARS-CoV-2 S protein is increasingly becoming more positively charged as new variants emerge, favoring its binding affinity to ACE2 and potentially also heparan sulfate and heparin [24,25]. This suggests that the use of heparin may provide strain-agnostic protection against SARS-CoV-2 infection. 

### 3.2. Inhibition of SARS-CoV-2 Replication by Heparin in Vero Cells

To study the effects of heparin on SARS-CoV-2 infection, we initially performed several experiments using permissive Vero cells. Culture of Vero cells with up to 250 μg/mL of heparin did not affect cell viability (Appendix A). At a MOI of 0.0002 (10^2^ TCID_50_/well), cells cultured in the presence of 250 μg/mL of heparin had approximately 10-fold less infectious virus in the supernatant at 3 days post infection (dpi) than untreated cells (Figure 2A). No significant difference in virus titers was observed in cells treated with 125 μg/mL of heparin compared to untreated cells at 3 dpi (Figure 2A). Heparin treatment did not inhibit virus titers if cells were infected with a higher MOI of 0.02 (Appendix A. These data confirmed that heparin treatment inhibited virus replication in a dose-dependent manner, where both the dose of the virus and the heparin dose were important.

To determine whether pretreatment of virus inoculum alone would be sufficient to inhibit infection of Vero cells, we pretreated 10^2^ TCID_50_ of virus with 250 μg of heparin for 1 h prior to infection, and after viral inoculation, we cultured the Vero cells either in the presence or absence of heparin. While culture in the presence of heparin inhibited SARS-CoV-2 replication by 10-fold as observed previously, pretreatment of the inoculum only was not sufficient to inhibit virus replication (Figure 2B). At 3 dpi, no cytopathic effect following SARS-CoV-2 infection was observed in cells cultured in the presence of heparin (Figure 2C). In contrast, a cytopathic effect was observed in untreated cells and cells infected with pretreated virus (Figure 2C), indicating that the continued presence of heparin is required for inhibition of virus growth in Vero cells.

To confirm inhibition of virus replication, we also quantified viral genomes by RT-PCR. The presence of heparin in supernatants from heparin-treated cells inhibited viral genome amplification by RT-PCR (Appendix A). Heparin is known to interact with DNA and can inhibit RT-PCR analysis of samples [26,27]. To overcome the inhibitory effect of heparin in RT-PCR, we extracted RNA from various concentrations of SARS-CoV-2 virus in the presence or absence of heparin and pretreated with heparinase I or a control buffer before RT-PCR analysis. Using this method, we showed a 100% rescue of Ct values (Appendix A). As observed for infectious virus titers, cells cultured in the presence of 250 μg/mL of heparin had approximately 10- to 100-fold fewer viral genomes in the supernatant up to 3 dpi than untreated cells (Figure 2D). 

To assess intracellular virus, we infected Vero cells with a GFP-expressing SARS-CoV-2 reporter virus and analyzed GFP expression by flow cytometry and microscopy (Appendix A). Significantly fewer GFP+ cells were observed in heparin-treated samples at 1 dpi, as determined by flow cytometry (Figure 2E,F). No clear difference was observed at 3 dpi (Figure 2F). However, this result is likely affected by the cell death observed at this timepoint in untreated virus-infected cells (Figure 2C). In support of these data, we also observed fewer GFP+ cells by microscopy when infection occurred in the presence of heparin (Figure 2G). Together, these data demonstrate that heparin can inhibit SARS-CoV-2 infection in Vero cells. 

### 3.3. Inhibition of SARS-CoV-2 Replication by Heparin in Human Nasal Epithelial Cells

To determine whether intranasal treatment with heparin is likely to be effective against SARS-CoV-2 infection of the upper respiratory tract, we assessed viral replication in human nasal epithelial cells treated with heparin. In cells obtained from three human donors, we observed an approximately 1000-fold decrease in infectious virus by 5 dpi when cells were cultured in the presence of either 125 μg/mL or 250 μg/mL of heparin (Figure 3A). Variability was observed in virus growth over time between the three male donors, particularly between the younger (ages 33, 35) and the older donors (age 67). While variability has been observed in similar models irrespective of donor age [28], some effects of human ALI cultures due to donor age and SARS-CoV-2 receptor expression have been reported previously [29,30]. Due to the variability of virus growth kinetics between donors, we also expressed the results as percentage inhibition relative to untreated SARS-CoV-2-infected cells from the same donor and saw nearly 100% inhibition of virus replication on 3, 5 and 7 dpi in the presence of heparin (Figure 3B). Although inhibition was observed at 1 dpi with both doses of heparin, it was more evident at 250 μg/mL of heparin. In contrast, a trend toward inhibition was observed with 125 μg/mL and not 250 μg/mL of heparin at 0 dpi. Both differences were modest as measured by virus titer (TCID_50_) (Figure 3A). A similar trend for reduced viral genomes was observed at 4 and 6 dpi, correlating with an 80% reduction in viral genome levels at 4 dpi (Figure 3C,D). There was more inconsistency in the inhibition of viral genomes relative to infectious virus, suggesting that there were potentially higher levels of noninfectious virus present in the supernatant. We observed no consistent difference in virus growth when the virus inoculum only was pretreated with heparin prior to infection (Appendix A).

To confirm that intracellular virus was also decreased by heparin treatment, we stained human nasal epithelial cells for dsRNA and SARS-CoV-2 nucleocapsid at 7 dpi. While infected cells were observed in the untreated nasal epithelial cells, we did not find dsRNA or SARS-CoV-2 nucleocapsid stained cells in heparin-treated nasal epithelial cells (Figure 3E). Together, these data indicate that heparin treatment inhibits SARS-CoV-2 infection of differentiated human nasal epithelial cells.

## 4. Discussion

Here, we show that heparin can effectively inhibit virus growth in both Vero cells and human nasal epithelial cells using several complementary approaches. Infectious virus was reduced approximately 10- to 100-fold in heparin-treated Vero cells compared to untreated cells, correlating with a similar reduction in viral genomes, reduced intracellular virus and prevention of a cytopathic effect. In human nasal epithelial cells, an approximate 1000-fold reduction in infectious virus was observed in the presence of heparin, associated with a modest reduction in viral genomes and the absence of detectable intracellular virus.

The degree of inhibition of virus replication was greater in the nasal epithelium compared to Vero cells. Furthermore, robust inhibition was observed with both doses of heparin in the nasal epithelium but only sustained with the higher dose in Vero cells. This difference may be related to the susceptibility of each cell type to SARS-CoV-2 infection, as we observed that nasal epithelial cells require a higher MOI (0.2) than Vero cells (0.0002) to establish virus growth. We also observed no inhibition of virus growth in Vero cells infected with an MOI of 0.02 and treated with 250 μg/mL of heparin. This suggests that, in addition to heparin dose, virus inoculum dose affects the antiviral success of heparin. While it is important to consider that this effect was only shown in Vero cells that are highly susceptible to SARS-CoV-2 infection, it stresses the importance of using heparin as a prophylaxis at the site of the initial point of infection when virus levels are likely to be at their lowest. Therefore, it is encouraging that such robust antiviral activity was observed in the nasal epithelium, given the physiological relevance of these cells to human infection and prophylaxis.

The INHERIT (Intranasal Heparin Treatment To Reduce Transmission Among Household Contacts of COVID-19 Positive Adults and Children) study is a clinical trial based in Australia (ClinicalTrials.gov Identifier: NCT05204550), with planned enrollment to begin this year, that will analyze the protection afforded to household contacts by intranasal heparin. In this study, following the detection of a SARS-CoV-2 infection index case, household contacts will self-administer heparin three times a day over a period of 10 days (8400 IU/daily). SARS-CoV-2 positivity, as determined by qPCR, and the development of symptomatic COVID-19 over the following 10 days will be assessed in household contacts. This study will determine whether intranasal heparin therapy has any prophylactic efficacy against SARS-CoV-2. 

Our data confirm that with this type of targeted approach to heparin-treatment, the nasal epithelium may be a successful target for antiviral therapy. However, we have uncovered some considerations for the INHERIT study and other clinical studies exploring the use of intranasal heparin. Firstly, we observed that inhibition of virus growth only occurs when cells are cultured in the continual presence of heparin. The concentration of heparin used in our study could be insufficient to bind and inhibit all the S protein or that the binding of heparin is not preferential to the binding to ACE2. In support of the latter, others have shown that the receptor-binding domain in the S protein binds with a significantly higher affinity to ACE2 than it does to heparin [31]. Thus, multiple doses of heparin spanning the period of exposure may be required to ensure adequate protection against infection balanced against the rate of nasal clearance. It also suggests that heparan sulfate binding is less critical than ACE2 binding for SARS-CoV-2 infection, and therefore, heparin treatment is probably to be less effective than the blockade of ACE2 with monoclonal antibodies. However, in contrast with heparin, the expense of monoclonal antibody production will likely preclude its widespread use as a prophylaxis. The INHERIT study requires heparin use throughout the study. Our data suggests that this repeated delivery will have the best chance of effectiveness. We also observed an inhibitory effect of heparin on our ability to detect SARS-CoV-2 RNA by RT-PCR. To adequately detect viral RNA in heparin-treated samples, we employed an additional heparinase-treatment step. It is possible that nasal swabs collected from participants in any nasal heparin study may contain traces of heparin. Therefore, consideration of any inhibitory effect of heparin on RT-PCR detection of SARS-CoV-2 is critical to ensure study accuracy. Our method may prove useful in this endeavor. 

## 5. Conclusions

Overall, we have shown that heparin is effective at inhibiting SARS-CoV-2 replication in both cell lines and, importantly, the human nasal epithelium. These results support the continual analysis of heparin in clinical trials as a prophylaxis against SARS-CoV-2. 

## Figures and Tables

**Figure 1 viruses-14-02620-f001:**
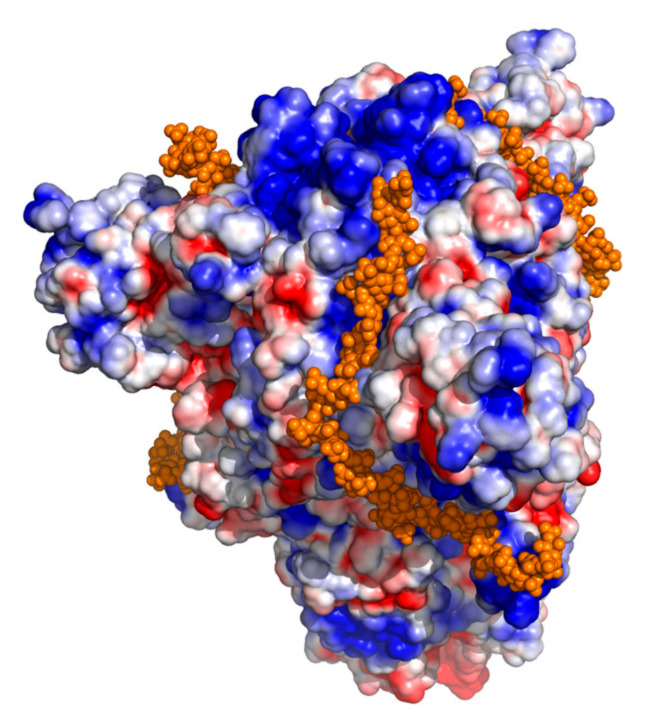
Model SARS-CoV-2 omicron variant spike protein—heparin complex. The solvent-accessible surface of the protein shown is colored by electrostatic charge, with negative regions blue and positive regions red. The modeled heparin is shown as orange spheres, with one strand of heparin bound to a groove in each of the trimeric spike protein chains.

**Figure 2 viruses-14-02620-f002:**
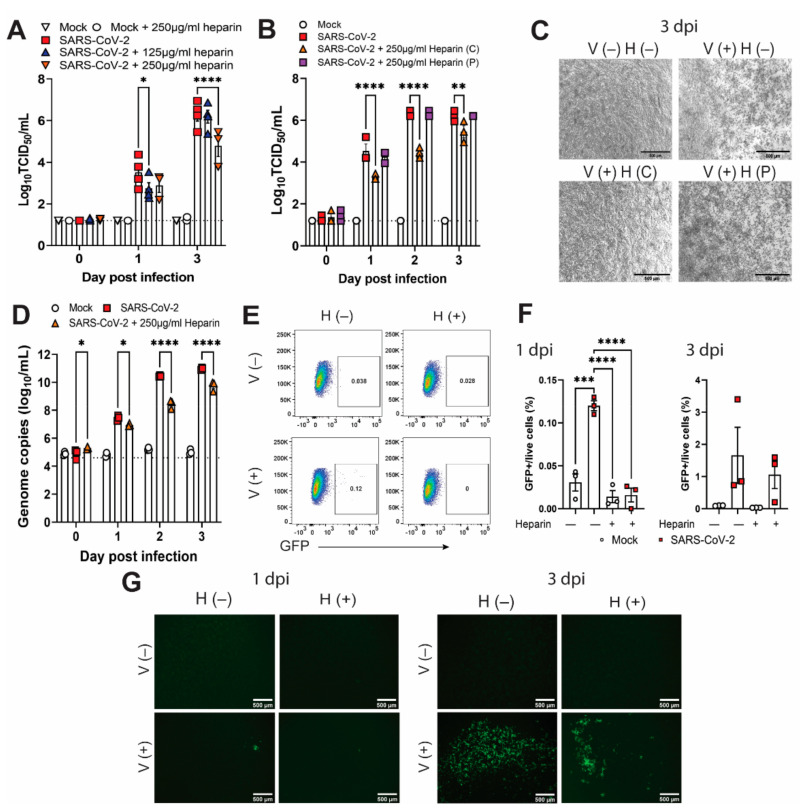
Heparin inhibits SARS-CoV-2 infection in Vero cells. (**A**) Infectious virus titers in supernatants from cells infected with SARS-CoV-2 (MOI 0.0002) and cultured with or without heparin for 3 days. (**B**) Infectious virus titers in supernatants from cells infected with virus incubated with or without heparin for 1 h prior to infection. Cells were cultured either with normal serum-free media (P = pretreatment only) or with continuous heparin (C = continuous) until day 3 post infection. (**C**) Microscopy images of cells at day 3 post infection from B. (**D**) Genome copies (E-gene) in supernatants from cells cultured, with or without heparin, for 3 days. (**E**) Representative flow cytometry plots from cells either mock infected (V−) or infected with GFP-expressing SARS-CoV-2 virus (V+) and cultured in the presence (H+) or absence (H−) of heparin. Percentage of GFP-expressing cells by flow cytometry (**F**) and microscopy images of GFP expression (**G**) on day 1 and 3 post infection from E. Data are a representative of 2 replicate experiments except for A that shows the mean titer +/− SEM of 3 replicates from 4 separate experiments. Scale bar = 500 μm. Log 10 transformed data were analyzed using Two-Way ANOVA (**A**,**B**,**D**) or One-Way ANOVA (**F**) with Dunnett’s multiple comparisons test. *, *p* < 0.05; **, *p* < 0.01; ***, *p* < 0.001 and ****, *p* < 0.0001.

**Figure 3 viruses-14-02620-f003:**
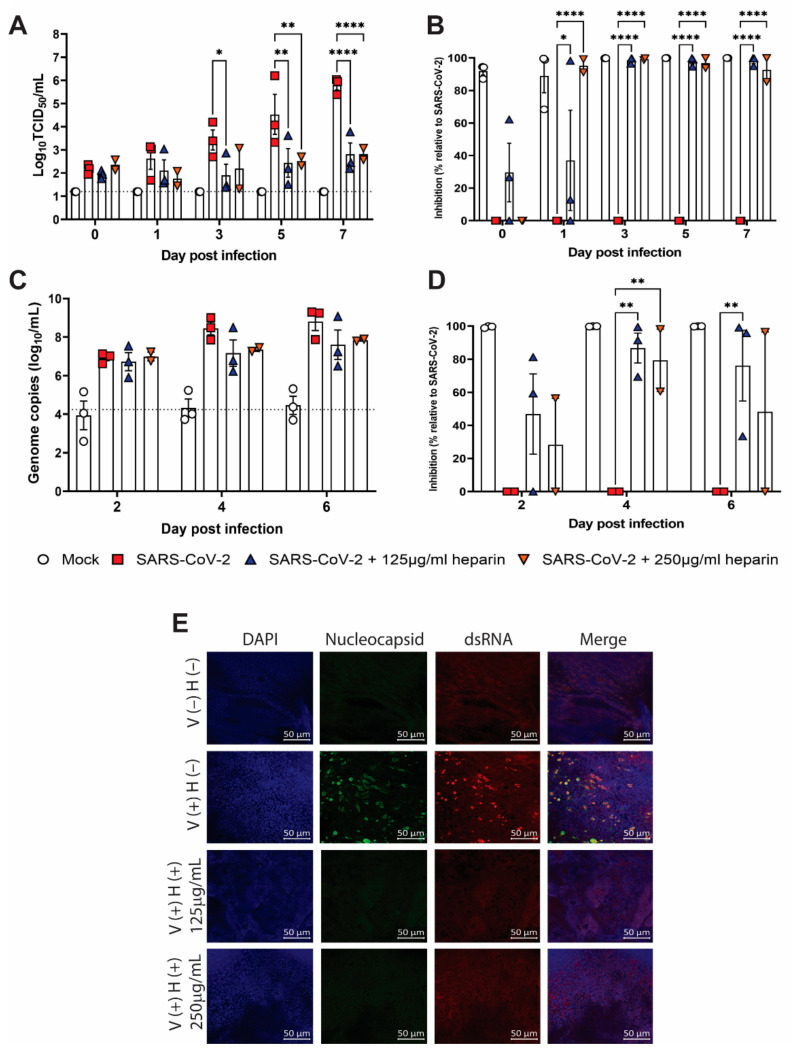
Heparin inhibits SARS-CoV-2 infection in human nasal epithelial cells. Infectious virus titers (**A**) and genome copies (**C**) on indicated days post infection in apical supernatants from cells cultured with or without heparin. Percent inhibition of virus titers (**B**) and genome copies (**D**) compared to SARS-CoV-2 infected cells cultured in the absence of heparin on the indicated days. Graphs show the mean +/− SEM of 2–4 replicates from 3 individual donors analyzed in 3 separate experiments, except for 250 μg/mL which was only tested in 2 donors. (**E**) Immunofluorescence images of cells at day 7 post infection stained for dsRNA (red), SARS-CoV-2 nucleocapsid (green) or DAPI (blue). Scale bar = 50 μM. Data were analyzed using a Two-Way ANOVA with Dunnett’s multiple comparisons test. *, *p* < 0.05; **, *p* > 0.01 and ****, *p* < 0.0001.

## Data Availability

Data will be made available upon request.

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
