# Peer review of "Heparin Inhibits SARS-CoV-2 Replication in Human Nasal Epithelial Cells"

_viruses, 2022, doi:10.3390/v14122620_

Round 1

Reviewer 1 Report

In this manuscript, the authors present their findings on the antiviral efficacy of Heparin in primary human nasal epithelial cells. The manuscript is well written and logically presented with adequate descriptions of the methods employed. The results are also clearly presented with the exception of figure 2, where more detail is required to describe panel E. In this case, the axes are undefined (what is V+ vs. V-) and the font is too small to read. In Figure 3, panel B, there was no discussion regarding the significant reduction in SARS-CoV-2 RNA when treated with 125ug/ml heparin but not 250 at time 0.

As acknowledged by the authors in the introduction, the antiviral efficacy of heparin and/or its derivatives have been reported previously in tissue culture including Vero cells and we learn in the discussion that a clinical trial is ongoing potentially diminishing the impact of these studies. However, the use of primary nasal epithelial cells is novel and those data along with observations regarding methods to remove heparin for RT-PCR is valuable, though those studies and the discussion are minimal. The authors elude to significant variability in infectivity of SARS-CoV-2 between donors which is interesting given the differential outcomes in people and especially between sexes (acknowledging that cells were only collected from one female donor). Additional detail on these differences and implications for therapeutic interventions would add to the novelty of the manuscript.

Other items to consider:

Line 197: washed three times instead of wash

Line 293: inoculum was only instead of was only.

Author Response

In this manuscript, the authors present their findings on the antiviral efficacy of Heparin in primary human nasal epithelial cells. The manuscript is well written and logically presented with adequate descriptions of the methods employed.

  1. The results are also clearly presented with the exception of figure 2, where more detail is required to describe panel E. In this case, the axes are undefined (what is V+ vs. V-) and the font is too small to read.

Response: We thank the reviewer for the comment.

Additional information related to panel E is now included in the figure legend as follows:

(E) Representative flow cytometry plots from cells either mock infected (V-) or infected with GFP-expressing SARS-CoV-2 virus (V+) and cultured in the presence (H+) or absence (H-) of heparin.

The font size on the flow cytometry plots in Figure 2 and Supplementary Figure 1 have been increased.

  1. In Figure 3, panel B, there was no discussion regarding the significant reduction in SARS-CoV-2 RNA when treated with 125ug/ml heparin but not 250 at time 0.

Response: As per Reviewer 3’s suggestions, figure 3 panel B was re-plotted to show percentage inhibition relative to SARS-CoV-2 infection (i.e SARS-CoV-2 infection is now shown as 0% inhibition). In addition, we performed a two-way ANOVA on these data instead of one-way ANOVA to consider the variability across days in addition to between culture conditions. In doing so, the 0 dpi results are no longer significantly different. However, we acknowledge that there is a trend for inhibition with 125μg/ml heparin but not the 250μg/ml heparin samples in panel B, even though the difference in the raw virus titre (TCID50) values in panel A is modest.

We have added the following sentence to the results (lines 301-304):

Although inhibition was observed at 1 dpi with both doses of heparin, it was more evident with 250 μg/mL of heparin. In contrast, a trend for inhibition was observed with 125 μg/mL and not 250 μg/mL of heparin at 0 dpi. Both differences were modest as measured by virus titre (TCID50) (Fig. 3a).   

  1. As acknowledged by the authors in the introduction, the antiviral efficacy of heparin and/or its derivatives have been reported previously in tissue culture including Vero cells and we learn in the discussion that a clinical trial is ongoing potentially diminishing the impact of these studies. However, the use of primary nasal epithelial cells is novel and those data along with observations regarding methods to remove heparin for RT-PCR is valuable, though those studies and the discussion are minimal. The authors elude to significant variability in infectivity of SARS-CoV-2 between donors which is interesting given the differential outcomes in people and especially between sexes (acknowledging that cells were only collected from one female donor). Additional detail on these differences and implications for therapeutic interventions would add to the novelty of the manuscript.

Response: We thank the reviewer for this suggestion and agree that it would be interesting to determine whether there is a sex-specific difference in the response to heparin. The experiments shown in Figure 3 were completed using 3 male donors aged 33, 35 and 67 years. Supplementary Figure 1c was completed with 1 female donor aged 26. Therefore, the variability between donors in Figure 3 is unlikely to be attributed to sex. Interestingly, it was in the cells from the 67-year-old donor that we observed slightly delayed virus growth over time. Therefore, it is possible that age may be a contributing factor to the observed variability. Age-related differences in SARS-CoV-2 replication have been documented by other studies.

To address these issues, we have added the following sentences to the results section (lines 294-298):

Variability was observed in virus growth over time between the three male donors, particularly between the younger (age 33,35) and the older donor (age 67). While variability has been observed in similar models irrespective of donor age [28], some effects of human ALI cultures due to donor age and SARS-CoV-2 receptor expression have been reported previously [29, 30].

Other items to consider:

  1. Line 197: washed three times instead of wash

This error has been amended

  1. Line 293: inoculum was only instead of was only.

This error has been amended

Reviewer 2 Report

In this manuscript, Yang Lee et al. show that heparin can bind to the spike protein of SARS-CoV-2 to inhibit SARS-CoV-2 infection in human nasal cells. This finding was already reported previously by other studies such as  PMID: 33368089, PMID: 35060381 and PMID: 34929169, among others. This topic has even been reviewed in PMID: 33569392. Therefore, the present manuscript has a strong lack of novelty, but it can be used as secondary confirmation of previous finding. Still, the manuscript is well-written and scientifically sounding. Other comments can be found below:

- Materials and methods, there is no indication about which SARS-CoV-2 strain was used for the experiments. SARS-CoV-2 strain is an important factor since it can influence disease outcome and in vitro infectivity.

- Lines 234-236, authors state that infection at higher MOI could not be inhibited by heparin. This fact points towards a prophylactic use of heparin instead rather than a treatment once disease manifests itself in patients because a higher MOI would be related with severe ongoing disease. This should be mentioned in Discussion.

- Figure 2G / 3H - scale bar can not be read with the current size and quality.

- Was confocal or epifluorescence microscopy used to perform immunofluorescence? DAPI channel looks like background signal, more consistent with epifluorescence.

- Discussion, lines 334-341, on top of this I would add that pre-treatment of SARS-CoV-2 with heparin was not enough to inhibit infection, which suggests that heparin binding to ACE2 might be more important to inhibit infection.

Author Response

In this manuscript, Yang Lee et al. show that heparin can bind to the spike protein of SARS-CoV-2 to inhibit SARS-CoV-2 infection in human nasal cells. This finding was already reported previously by other studies such as  PMID: 33368089, PMID: 35060381 and PMID: 34929169, among others. This topic has even been reviewed in PMID: 33569392. Therefore, the present manuscript has a strong lack of novelty, but it can be used as secondary confirmation of previous finding. Still, the manuscript is well-written and scientifically sounding. Other comments can be found below:

1. Materials and methods, there is no indication about which SARS-CoV-2 strain was used for the experiments. SARS-CoV-2 strain is an important factor since it can influence disease outcome and in vitro

Response: The following two viruses were used for this study:

  1. The ancestral SARS-CoV-2 (Australia/VIC01/2020).
  2. GFP-tagged SARS-CoV-2 (icSARS-CoV-2-GFP/WA1)

This information is given on lines 115-119.

2. Lines 234-236, authors state that infection at higher MOI could not be inhibited by heparin. This fact points towards a prophylactic use of heparin instead rather than a treatment once disease manifests itself in patients because a higher MOI would be related with severe ongoing disease. This should be mentioned in Discussion.

Response: We thank the reviewer for this interesting discussion point and agree that it suggests that the use of heparin would be most beneficial for prophylaxis.

We have added a paragraph in the discussion (Lines 335-348) related to this:

The degree of inhibition of virus replication was greater in the nasal epithelium compared to Vero cells. Furthermore, robust inhibition was observed with both doses of heparin in nasal epithelium but only sustained with the higher dose in Vero cells. This difference may be related to susceptibility of each cell type to SARS-CoV-2 infection as we observed that nasal epithelial cells require a higher MOI (0.2) than Vero cells (0.0002) to establish virus growth. We also observed no inhibition of virus growth in Vero cells infected with an MOI of 0.02 and treated with 250 μg/mL of heparin. This suggests that in addition to heparin dose, virus inoculum dose affects the antiviral success of heparin. While it is important to consider than this effect was only shown in a Vero cells that are highly susceptible to SARS-CoV-2 infection, it stresses the importance of using heparin as a prophylaxis at the site of initial point of infection when virus levels are likely to be at their lowest. Therefore, it is encouraging that such robust antiviral activity was observed in the nasal epithelium given the physiological relevance of these cells to human infection and prophylaxis.

3. Figure 2G / 3H - scale bar can not be read with the current size and quality.

Response: The scale bar width and text size on both these panels have been increased.

4. Was confocal or epifluorescence microscopy used to perform immunofluorescence? DAPI channel looks like background signal, more consistent with epifluorescence.

Response: These images were obtained using a confocal microscope (LSM 780). We recognise that it is difficult to observe distinct individual cells in these images, particularly for the DAPI expression because the ALI cultures have many layers of cells. For this imaging we cut out the fixed transwell membranes on which the nasal epithelial cells were grown and performed whole mount staining with the images looking down onto the multiple layers of ALI cells with stacks of DAPI stained cells overlapping each other that are not in focus in the same plane.

We have now mentioned that this is whole mount staining in the methods section and added additional information (lines 198-210) as follows:

Nasal epithelium on transwells were fixed with 4% formaldehyde solution for 30 minutes, washed three times with PBS and stored at 4ºC until staining. Transwells were whole mount stained by treating with blocking solution (5% normal goat serum, 0.1% Triton-X in PBS) for 1 hour at room temperature and staining overnight at 4ºC with mouse anti-dsRNA (J2, Mouse IgG2a kappa, Australian Biosearch) and rabbit anti-SARS coronavirus nucleocapsid polyclonal antibody (PA1-41098, Thermo Fisher Scientific). After washing with PBS, primary antibodies were detected using Goat anti-Mouse IgG (H+L) Cross-Adsorbed Alexa Fluor 647 (Thermo Fisher Scientific) and Goat anti-Rabbit IgG (H+L) Cross-Adsorbed Alexa Fluor 568 (Thermo Fisher Scientific) for 1 hour at room temperature. Cells were counterstained with DAPI (Thermo Fisher Scientific) and transwells mounted with VECTASHIELD Antifade Mounting Medium (Vector Laboratories) on SuperFrost slides (ProSciTech) before visualization using the LSM 780 (Ziess).

5. Discussion, lines 334-341, on top of this I would add that pre-treatment of SARS-CoV-2 with heparin was not enough to inhibit infection, which suggests that heparin binding to ACE2 might be more important to inhibit infection.

Response: We have interpreted the reviewer’s statement to mean that blockade of ACE2 might be a better inhibitor of virus replication than blockade of heparan sulphate since, unlike blockade of ACE2, heparin pre-treatment alone is not enough to inhibit infection. We agree with this point and have added the following sentences into the discussion (Lines 369-373):

It also suggests that heparan sulfate binding is less critical than ACE2 binding for SARS-CoV-2 infection and therefore heparin treatment is likely to be less effective than blockade of ACE2 with monoclonal antibodies. However, in contrast with heparin, the expense of monoclonal antibody production will likely preclude its widespread use as a prophylaxis.

Reviewer 3 Report

The manuscript "Heparin inhibits SARS-CoV-2 replication in human nasal epithelial cells" by Lee et al., is well written. The authors study the antiviral effect of heparin on SARS-CoV-2, however, there are major flaws in the experimental setup and data analysis.

1. The approach has no clinical significance as heparin is not at all protective at even MOI 0.002. 

2. How are the authors removing the heparin solution from the virus after incubation for 1 hour?

3. The authors want to block viral entry with heparin but there are no assays to measure the viral entry and estimation of viral RNA in the cells.

4. The number of data points used for analysis in Fig. 3B and 3D is not significant and Fig.3D has a huge error bar which shows the inconsistency of the results.

5. The graphs need to be reanalyzed as SARS-CoV-2 alone has been shown to have 100% inhibition.

6. The authors' conclusion is not supported by results that heparin effectively inhibits virus growth as it works at only very low MOI and is not dose-dependent.

Author Response

The manuscript "Heparin inhibits SARS-CoV-2 replication in human nasal epithelial cells" by Lee et al., is well written. The authors study the antiviral effect of heparin on SARS-CoV-2, however, there are major flaws in the experimental setup and data analysis.

  1. The approach has no clinical significance as heparin is not at all protective at even MOI 0.002. 

Response: These studies were completed in Vero cells, a line that is highly susceptible to SARS-CoV-2 infection even at extremely low MOIs. This is in stark contrast with the human nasal epithelium which requires a much higher MOI to establish infection. Indeed, we were unable to infect nasal epithelial cells with 102 of SARS-CoV-2 (data not shown). However, the use of primary human nasal epithelial cells (hNECs) is a much better approximation than Vero cells, of what would be occurring in people, particularly in prophylaxis. In hNECs, the doses of heparin were very effective at inhibiting virus infection. Therefore, while Vero cells are a useful cell culture model due to their tractability, experiments in human nasal epithelial cells are more physiologically relevant.

We have added an additional paragraph in the discussion (Lines 335-348) as follows:

The degree of inhibition of virus replication was greater in the nasal epithelium compared to Vero cells. Furthermore, robust inhibition was observed with both doses of heparin in nasal epithelium but only sustained with the higher dose in Vero cells. This difference may be related to susceptibility of each cell type to SARS-CoV-2 infection as we observed that nasal epithelial cells require a higher MOI (0.2) than Vero cells (0.0002) to establish virus growth. We also observed no inhibition of virus growth in Vero cells infected with an MOI of 0.02 and treated with 250 μg/mL of heparin. This suggests that in addition to heparin dose, virus inoculum dose affects the antiviral success of heparin. While it is important to consider than this effect was only shown in a Vero cells that are highly susceptible to SARS-CoV-2 infection, it stresses the importance of using heparin as a prophylaxis at the site of initial point of infection when virus levels are likely to be at their lowest. Therefore, it is encouraging that such robust antiviral activity was observed in the nasal epithelium given the physiological relevance of these cells to human infection and prophylaxis.

  1. How are the authors removing the heparin solution from the virus after incubation for 1 hour?

Response: Following incubation of the virus inoculum/heparin mix with the cells, the inoculum was aspirated. The cells were then washed with media or PBS once for Vero cells and 3 times for nasal epithelial cells. Fresh media was added to the Vero cells for culture for an additional 3 days with media collected and replaced daily. Nasal epithelial cells were cultured for 7 days with no media on the apical surface and basal media changed every 2 days.

The following additional details have been added to the methods section (Lines 167-169):

Virus/heparin solution was removed from the cells by washing with either serum-free MEM (Vero) or PBS (ALI cells) and cells were cultured with or without heparin for the remainder of the experiment.

  1. The authors want to block viral entry with heparin but there are no assays to measure the viral entry and estimation of viral RNA in the cells.

Response: We estimated intracellular virus levels by flow cytometry using a GFP-expressing virus and by immunofluorescence using antibodies to detect dsRNA and viral protein. However, we acknowledge that these experiments were done at least 24 hours post-infection and therefore are not a strict read-out of initial virus entry. For this reason, we have referred to all these experiments as a measure of intracellular virus to confirm the data obtained from measuring extracellular virus in the supernatant.

We have made no additional changes related to this point.

  1. The number of data points used for analysis in Fig. 3B and 3D is not significant and Fig.3D has a huge error bar which shows the inconsistency of the results.

Response: For experiments shown in Figure 3 we used 3 individual donors representing 3 independent experiments each with 2-4 replicates per condition. We then calculated mean values per experiment and show a composite graph. We agree that there is significant variability between donors, particularly for the RNA analysis. Our interpretation is that there may be non-infectious viral RNA in the supernatant that is detected by RT-PCR. Therefore, we have:

-Added additional information related to the experimental design into the figure legend as follows:

Graphs show the mean +/- SEM of 2 to 4 replicates from 3 individual donors analyzed in 3 separate experiments.

-Performed Two-Way ANOVA instead of One-Way ANOVA for a more robust analysis of virus titres relative to both time and culture condition

-Included an additional discussion on RNA genome levels (see Lines 306-308)

-Included an additional discussion on experimental variability (see Lines 294-298)

  1. The graphs need to be reanalyzed as SARS-CoV-2 alone has been shown to have 100% inhibition.

Response: We thank the reviewer for the suggestion. The graphs have been amended to display untreated virus infected cells as 0% inhibition.

  1. The authors' conclusion is not supported by results that heparin effectively inhibits virus growth as it works at only very low MOI and is not dose-dependent.

Response: We respectfully disagree with this statement. We have shown that there is a significant reduction in virus titre in Vero cells measured by infectivity assay, RT-PCR, flow cytometry and microscopy with a reduction in cytopathic effect. We have shown that 250 μg/mL of heparin induces better inhibition than 125 μg/mL of heparin and that increasing the virus MOI 100-fold leads to a loss of anti-viral activity. Thus, inhibition is dependent on virus-dose and heparin-dose. Importantly, we show there is even better inhibition in physiologically relevant human nasal epithelial cells than in Vero cells. These cells are less susceptible to SARS-CoV-2 infection than Vero cells and heparin treatment more accurately represents prophylaxis at the site of initial infection in humans. In hNECs from 3 separate donors, we showed almost complete inhibition of infectious virus production. This correlated with a reduction in viral genome copies and complete absence of intracellular virus (GFP) by 7 dpi. These data in 2 cell types with multiple techniques clearly show that heparin inhibits SARS-CoV-2 replication. 

Round 2

Reviewer 3 Report

The authors have considerately improved the manuscript. It has been mentioned that the  Graphs show the mean +/- SEM of 2 to 4 replicates from 3 individual donors analyzed in 3 separate experiments, however, there are only two or three spots on the bar graph. How is the revision justified in the figure 3?

Author Response

The authors have considerately improved the manuscript. It has been mentioned that the  Graphs show the mean +/- SEM of 2 to 4 replicates from 3 individual donors analyzed in 3 separate experiments, however, there are only two or three spots on the bar graph. How is the revision justified in the figure 3?

We are sorry that this was not clear in the figure legend. For one of the donors we only tested the 125 ug/ml dose. Therefore for all groups in this figure n=3 except for the 250 ug/ml group where n=2. We have added the following addition to the sentence in the figure 3 legend :

Graphs show the mean +/- SEM of 2 to 4 replicates from 3 individual donors analyzed in 3 separate experiments, except for 250 ug/mL that was only tested in 2 donors.